# Combined Immunotherapy with Chemotherapy versus Bevacizumab with Chemotherapy in First-Line Treatment of Driver-Gene-Negative Non-Squamous Non-Small Cell Lung Cancer: An Updated Systematic Review and Network Meta-Analysis

**DOI:** 10.3390/jcm11061655

**Published:** 2022-03-16

**Authors:** Yue Chai, Xinyu Wu, Hua Bai, Jianchun Duan

**Affiliations:** Department of Medical Oncology, National Cancer Center/National Clinical Research Center for Cancer/Cancer Hospital, Chinese Academy of Medical Sciences and Peking Union Medical College, Beijing 100021, China; cy972628990@163.com (Y.C.); wxy_9707@163.com (X.W.); baihuahb@sina.com (H.B.)

**Keywords:** anti-angiogenic therapy, immune checkpoint inhibitor, non-squamous non-small cell lung cancer, network meta-analysis

## Abstract

Background: A network meta-analysis was conducted to summarize randomized control trials and updated results to evaluate the efficacy and safety profiles of existing first-line therapies for advanced non-squamous non-small cell lung cancer (NSCLC) patients without known driver gene mutations. Patients and Methods: Eligible studies were identified following a systematic search of the Cochrane Library, PubMed, Embase, Web of Science, Wanfang Data, and the China Knowledge Resource Integrated Database from January 2000 to December 2021. Results: Nineteen trials involving 8176 patients with driver-gene-negative advanced non-squamous NSCLC were included. For patients with driver-gene-negative advanced NSCLC, immunotherapy + chemotherapy (IC) significantly prolonged overall survival (OS) (hazard ratio (HR), 0.80; 95% confidence intervals (CI): 0.67–0.95) and progression-free survival (PFS) (HR, 0.68; 95% CI: 0.53–0.86) compared with bevacizumab + chemotherapy (BC), with a similar objective response rate and incidence of ≥3 treatment-related adverse events (TRAEs) (risk ratios (RR), 0.98; 95% CI: 0.79–1.21/RR, 0.89; 95% CI: 0.61–1.28; respectively) compared with BC. IC yielded a superior PFS rate (HR, 1.59; 95% CI: 1.05–2.38) compared to BC in the subgroup of patients < 65 years old. Conclusions: Currently, IC is a more efficient first-line therapy for driver-gene-negative advanced non-squamous NSCLC patients, with prolonged PFS and OS, as well as a comparatively lower risk of ≥3 TRAEs compared to BC.

## 1. Introduction

Lung cancer is one of the dominant lethal malignancies in the world. Non-small cell lung cancer (NSCLC) accounts for nearly 85% of lung cancer [1]. Common driver genes of NSCLC include epidermal growth factor receptor (EGFR), anaplastic lymphoma kinase (ALK), the proto-oncogene 1 (ROS), receptor tyrosine kinase (ROS1), B-Raf proto-oncogene, serine/threonine kinase (BRAF), MET proto-oncogene, receptor tyrosine kinase (MET) (exon 14 skipping mutation), neurotrophic receptor tyrosine kinase 2 (NTRK), Ret proto-oncogene (RET), etc., which were defined in the National Comprehensive Cancer Network (NCCN) guidelines of NSCLC in 2021. The most prominent driver gene mutations were sensitive EGFR mutations, followed by ALK rearrangements. For decades, the standard first-line regimen for advanced NSCLC with no actionable driver gene mutations was platinum-based chemotherapy (CT). The survival outcome has been greatly improved with immunotherapy as a newly introduced treatment into the regimen for driver-gene-negative advanced NSCLC. Multiple studies have proved that patients with driver-gene-negative advanced NSCLC with programmed death ligand-1 (PD-L1) ≥ 50% using single-agent immunotherapy have a significantly better outcome than standard CT [2,3,4,5]. In 2006, bevacizumab (BEV) was approved by the U.S. Food and Drug Administration (FDA) as the first-line treatment of advanced or metastatic non-squamous NSCLC [6]. For patients with a comparatively lower expression of PD-L1, immune checkpoint inhibitors (ICIs) or BEV combined with CT were suggested as a preferred recommendation for patients with driver-gene-negative advanced non-squamous NSCLC without corresponding contraindications. With the advent of the age of immunotherapy, a breakthrough has been brought to the treatment of NSCLC. In recent years, ICIs targeting programmed death 1 (PD-1), PD-L1, and cytotoxic T lymphocyte-associated antigen-4 (CTLA-4) as the first-line treatment for driver-gene-negative advanced NSCLC are a hot research territory. In 2016, the FDA approved PD-1/PD-L1 inhibitors, either administered as a monotherapy or in combination, for first-line or second-line treatment of patients with driver-gene-negative advanced NSCLC [7,8]. Carrelizumab, sintilizumab, and tislelizumab have been China Food and Drug Administration (CFDA)-approved successively for driver-gene-negative advanced non-squamous NSCLC. Interim study results of CHOICE-01 were reported at the 2021 World Conference on Lung Cancer (WCLC). This study showed that compared with CT alone, toripalimab plus CT in the first-line treatment of driver-gene-negative advanced NSCLC could significantly prolong the PFS and reduce the risk of disease progression in patients with driver-gene-negative advanced NSCLC. In addition, a benefit trend toward improved overall survival (OS) was observed. In May 2020, nivolumab combined with ipilimumab for the treatment of patients with advanced, drive-gene-negative, PD-L1 ≥ 1% NSCLC was approved by the FDA. Without a doubt, the era of immunotherapy for driver-gene-negative NSCLC is coming.

However, few studies directly compared the efficacy and safety of CT with immunotherapy or anti-angiogenetic agents in the first-line treatment for patients with driver-gene-negative advanced non-squamous NSCLC. A network meta-analysis (NMA) was conducted to preliminarily summarize the randomized control trials (RCTs) and updated results to assess the efficacy and safety profiles of existing first-line therapies for driver-gene-negative advanced non-squamous NSCLC patients.

## 2. Material and Methods

### 2.1. Literature Search and Selection

Eligible studies were identified following the systematic search of the Cochrane Library, PubMed, Embase, Web of Science, Wanfang Data, and the China Knowledge Resource Integrated Database from January 2000 to December 2021. The language was restricted to English. The following terms were used: “non-small-cell lung cancer”, “non-squamous”, “lung cancer”, “PD-1 inhibitor”, “PD-L1 inhibitor”, “pembrolizumab”, “Keytruda”, “MK-3475”, “nivolumab”, “MDX-1106”, “ONO-4538”, “BMS-936558”, “Opdivo”, “atezolizumab”, “MPDL3280A”, “Tecentriq”, “RG7446”, “RG-7446”, “Durvalumab”, “Imfinzi”, “MEDI4736”, “Camrelizumab”, “SHR-1210”, “Tislelizumab”, “Sintilimab”, “IBI-308”, “CTLA 4 Antigen”, “Cytotoxic T Lymphocyte Associated Antigen 4”, “CD152 Antigen”, “CTLA-4 Protein”, “Cytotoxic T Lymphocyte Antigen 4”, “ONO-4538”, “MDX-1106”, “Ipilimumab”, “MDX-010”, “BMS-936558”, “CP-675”, “Tremelimumab”, “KN-046”, “Cadonilimab”, “ADG-116”, “Bevacizumab”, “Avastin”, “clinical trials”, “Randomized clinical trial”, and “phase”. The detailed search strategy is shown in Appendix A.

The inclusion criteria of eligible studies were as follows: (1) phase 2/3 randomized controlled clinical trials; (2) previously untreated NSCLC patients; (3) stage III B/IV according to TNM stage (AJCC version 7.0); (4) patients without known driver gene mutations (EGFR mutations or ALK rearrangements); (5) RCTs comparing an anti-angiogenic combined therapy to other treatment or an immunotherapy combined therapy to other treatment. The exclusion criteria were: (1) non-RCT studies; (2) patients not treated with first-line treatment; (3) data on disclosures from the same study at different times.

We followed the Preferred Reporting Items for Systematic Reviews and Meta-Analyses (PRISMA) checklist with the extension for NMA [9].

The full protocol is published on the PROSPERO website in December 2021 with the PROSPERO registration number CRD42021290355.

### 2.2. Data Extraction and Quality Assessment

Data extraction and cross-checking were undertaken independently by two investigators. The following information was then recorded in an Excel spreadsheet: the research number, the first author, the year of publication, the type of study design, the inclusion and exclusion criteria, the sample size of patients in each group, the follow-up time, and the objective response rate (ORR), based on the Response Evaluation Criteria in Solid Tumors (RECIST; criteria versions 1.0 and 1.1 according to the different publication years), progression-free survival (PFS), OS, and treatment-related adverse events, (TRAEs) based on the National Cancer Institute Common Toxicity Criteria for Adverse Events (versions 2.0, 3.0, and 4.0 according to the different publication years).

The potential risks of bias in trials were assessed independently by two researchers based on RCT’s Cochrane risk of bias assessment: (1) Method of generating random sequences; (2) allocation sequence concealment; (3) implementation of blinding; (4) the completion of results; (5) selective reporting assessment; (6) Other biases. These risks of bias were graded as three levels: low risk, high risk, and unclear risk. If there were disagreements, a third researcher was involved in the discussion to resolve them and explain the reasons (Appendix A).

### 2.3. Statistical Analysis

OS and PFS outcomes were expressed as hazard ratios (HR) with their 95% confidence interval (95% CI). ORR and TRAEs were calculated using the risk ratios (RR) and their 95% CI as a measure of association. A 95% CI, excluding 1, was considered statistically significant. In terms of PFS and OS, outcomes with HR < 1 would suggest better survival outcomes. ORR outcomes with RR > 1 would suggest better efficacies, whereas toxicity outcomes with RR < 1 would suggest better toxicity profiles.

This NMA was carried out to identify indirect comparisons between experimental groups. Risk of bias assessment was carried out using Review Manager (version 5.3 for Windows; Cochrane Collaboration, Oxford, UK). The network relation graphs in this NMA were conducted using STATA (Version 26.0; StataCorp LP, College Station, TX, USA). Network meta-analyses were performed using the GeMTC R package (version 4.1.2; R Foundation for Statistical Computing, Vienna, Austria). Random-effects models were conducted due to slightly different treatment modalities in each eligible trial. The parameters of the R were set as follows: 4 chains, 50,000 iterations, a thinning interval of 1 for OS and PFS, and a thinning interval of 10 for ORR and ≥3 TRAEs to minimize auto-correlation. Ranking probabilities of treatments were assessed utilizing the surface under the cumulative ranking (SUCRA) scores to show the likelihood of therapies in best-to-worst order (score of 0–1 and 1 is the best).

## 3. Results

### 3.1. Eligible Studies

A total of 896 trials were assessed for eligibility and 19 studies (8176 patients) met our inclusion criteria, of which 17 were phase three studies and two were phase two studies. Six trials (*n* = 2478) investigated BEV + CT (BC) (*n* = 1263) versus CT (*n* = 1215), nine trials (*n* = 4381) investigated immunotherapy + CT (IC) (*n* = 2501) versus CT (*n* = 1880), one trial (CheckMate 227 part 1A) investigated dual immunotherapy (DI) (*n* = 134) versus CT (*n* = 140), and one trial (CheckMate 9LA) investigated dual immunotherapy + CT (DIC) (*n* = 248) versus CT (*n* = 247). Only the IMpower 150 study investigated BEV + immunotherapy + CT (BIC) (n = 350) versus IC (*n* = 359) versus BC (*n* = 338). CheckMate 227 was divided into three parts, including part one A (patients with PD-L1 ≥ 1%, mainly compared DI with CT), part one B (patients with PD-L1 < 1%, mainly compared IC with CT), and part two (patients with PD-L1 < 1%, mainly compared IC and CT). Each part enrolled different patients. For ease of understanding, CheckMate 227 was regarded as three studies and all of them were enrolled in this NMA.

The study identification and selection process is demonstrated in Figure 1. The baseline characteristics and outcome measurements of the eligible trials are listed in Table 1.

A presentation of the network of OS, PFS, ORR, and ≥three TRAEs is provided in Figure 2. Indirect comparisons among IC, BC, BIC, DI, DIC, and CT are connected.

### 3.2. OS and PFS

Sixteen trials included a total of 7802 individual patients, where 2181 received IC, 1601 patients received BC, 350 patients received BIC, 248 patients received DIC, 278 patients received DI, and 3144 patients received CT, provided OS data. Nineteen trials included a total of 8535 individual patients, where 2806 received IC, 1601 patients received BC, 350 patients received BIC, 248 patients received DIC, 278 patients received DI, and 3481 patients received CT, provided PFS data.

IC had significantly prolonged OS (HR, 0.80; 95% CI: 0.67–0.95) and PFS (HR, 0.68; 95% CI: 0.53–0.86) compared with BC. BIC had significantly longer PFS (HR, 0.62; 95% CI: 0.41–0.95) but not OS (HR, 0.78; 95% CI: 0.58–1.04) compared with BC. BIC (HR, 0.70; 95% CI: 0.52–0.95/HR, 0.53; 95% CI: 0.34–0.86) and IC (HR, 0.73; 95% CI: 0.63–0.83/HR, 0.59; 95% CI: 0.51–0.68) had both longer OS and PFS than CT. DIC had significantly longer OS (HR, 0.69; 95% CI: 0.49–0.98) but not PFS (01.28, 0.95–1.76) compared with CT. BC and DI were statistically equivalent to CT for OS and PFS (Figure 3A). Forest plots of HRs for OS and PFS are shown in Appendix A.

### 3.3. ORR and Safety

Eighteen trials included a total of 7754 individual patients, where 2307 received IC, 1228 patients received BC, 350 patients received BIC, 246 patients received DIC, 278 patients received DI and 3345 patients received CT, provided ORR data. Among patients treated with BC with available data, the ORR was 39.0% (479/1228) compared to 50.3% (1161/2307) of patients who received IC (*p* < 0.01). Nine trials, including a total of 5079 individual patients, where 1929 received IC, 954 patients received BC, and 2196 patients received CT, provided necessary toxicity data.

In terms of ORR of first-line therapy, BIC showed significant efficacy compared with BC (RR, 1.33; 95% CI: 1.00–1.77), DIC (RR, 2.00; 95% CI: 1.25–3.22), and CT (RR, 2.27; 95% CI: 1.63–3.17). IC and BC was superior to DIC (RR, 1.47; 95% CI:1.03–2.11/RR, 1.50; 95% CI: 1.03–2.20) and CT (RR, 1.67; 95% CI:1.48–1.89/RR, 1.71; 95% CI: 1.44–2.02). DI (RR, 1.52; 95% CI: 1.03–2.24) was better than CT. DIC (RR, 1.14; 95% CI: 0.81–1.59) showed similar efficacy when compared with CT. IC (RR, 0.98; 95% CI: 0.79–1.21) showed similar efficacy compared to BC. When analyzing ≥3 treatment-related adverse events of these six treatment modalities, IC (RR, 1.24; 95% CI: 1.00–1.54) and BC (RR, 1.40; 95% CI: 1.04–1.89) were both significantly higher than CT. BC and IC showed similar incidences of ≥3 TRAEs (RR, 0.89; 95% CI: 0.61–1.28) (Figure 3B). Forest plots of RRs for ORR and ≥3 TRAEs are presented in Appendix A.

### 3.4. Subgroups of Various Clinicopathological Characteristics

In the Eastern Cooperative Oncology Group (ECOG) score = 0 subgroup, the PFS of IC was longer than that of CT (HR, 0.56; 95% CI: 0.42–0.75). In the ECOG score = 1, IC and BC both had significantly longer PFS (HR, 0.62; 95% CI: 0.50–0.74/HR, 0.42; 95% CI: 0.29–0.62) compared with CT. In smokers, IC (HR, 0.57; 95% CI: 0.42–0.72) and BC (HR, 0.51; 95% CI: 0.33–0.79) significantly prolonged PFS but not in OS compared with CT. In non-smokers, BC yielded superior PFS (HR, 0.39; 95% CI: 0.20–0.78) than CT. In the subgroup of patients < 65 years old, IC yielded superior PFS (HR, 1.59; 95% CI: 1.05–2.38) than BC, and IC and BC both had significantly longer PFS (HR, 0.63; 95% CI: 0.51–0.77/HR, 0.39; 95% CI: 0.28–0.57) compared with CT. In the subgroup of male patients, BIC and BC achieved significant OS (HR, 0.55; 95% CI: 0.31–0.97/HR, 0.69; 95% CI: 0.49–0.98) benefits compared with CT, while IC and BC achieved significant PFS (HR, 0.60; 95% CI: 0.47–0.79/HR, 0.45; 95% CI: 0.29–0.70) benefits compared with CT. In the subgroup of female patients, IC and BC achieved significant PFS (HR, 0.58; 95% CI: 0.45–0.75/HR, 0.45; 95% CI: 0.29–0.73) benefits, but not OS benefits compared with CT. In subgroups of patients ≥ 65 years old and patients with liver metastases, IC, BC, and CT all achieved equal outcomes on PFS and OS (Figure 4).

### 3.5. Rank Probabilities

The Bayesian ranking probabilities and corresponding SUCRA of the various interventions in different populations are shown in Appendix A. Treatments with the probability of being ranked first in OS are as follows: DIC (0.44), BIC (0.33), IC (0.14), DI (0.09), BC (0.00), CT (0.00). BIC was ranked the best therapy in terms of PFS and ORR with a probability of 0.63 and 0.91, respectively. BC was associated with the highest probability of ranking first for ≥3 TRAEs (0.73), followed by IC (0.27) and CT (0.00).

In the ECOG PS = 0/1, smokers/non-smokers, patients < 65 years old subgroups, treatments with the probability of being ranked first in OS are as follows: BIC, IC, BC, CT. In patients ≥ 65 years old, treatments with the probability of being ranked first in OS are as follows: BC, IC, CT. In the male subgroup, BIC ranked first in OS, followed by BC, IC, and CT. In the female subgroup, IC ranked first in OS, followed by BIC, BC, and CT. OS of BC ranked first, followed by IC and CT for patients with liver metastases. In terms of PFS, in the ECOG PS = 0/1, smokers/non-smokers, patients ≥ 65/< 65 years old subgroups, the male/female subgroup, patients with liver metastases, treatments with the probability of being ranked first are as follows: BC, IC, CT (Appendix A).

## 4. Discussion

As a result, this updated NMA enrolled 19 available studies including 8176 patients with driver-gene-negative advanced non-squamous NSCLC, which demonstrated the superior efficacy of IC as the first-line treatment for driver-gene-negative advanced non-squamous NSCLC compared to BC in the aspect of OS and PFS. Moreover, IC showed similar efficacy and incidences of ≥3 TRAEs compared with BC. Consequently, the present study provided a theoretical basis for selecting a more effective modality for patients with driver-gene-negative advanced non-squamous NSCLC.

Vascular endothelial growth factor (VEGF) expressed by cancer cells could promote angiogenesis, chemotaxis, and vasodilation, and further sustain tumor growth. Anti-angiogenetic drugs inhibit the combination of VEGF and vascular endothelial growth factor receptor (VEGFR), block the activation of downstream pathways, degrade the existing tumor vascular system and inhibit the formation of new blood vessels. Moreover, they also improve the efficacy of CT due to their anti-vascular permeability. The combination of BEV and CT in the treatment of NSCLC has received much attention in the era of CT. In phase 3 ECOG 4599 trial, 878 patients with treatment-naive advanced non-squamous NSCLC were treated with carboplatin and paclitaxel plus BEV or carboplatin and paclitaxel [10]. OS was significantly increased with BC compared to CT (median OS (mOS) 12.3 vs. 10.3 months, HR: 0.79, *p* = 0.003). PFS also increased (median PFS (mPFS) 6.2 vs. 4.5 months, HR: 0.66, *p* < 0.001) along with ORR of 35% and 15% (*p* < 0.001), respectively. Based on this research, the FDA approved BEV as the first-line treatment of advanced or metastatic non-squamous NSCLC in 2016 [6]. The results of the BEYOND study for Chinese patients with NSCLC were similar to those of the ECOG 4599 trial. The mOS of patients with NSCLC in the BEV combined with paclitaxel/carboplatin group was extended by 6.6 months and the mPFS was extended by 2.7 months, which indicated that BEV combined with paclitaxel/carboplatin in the treatment of Chinese patients with NSCLC was better than CT alone [11]. However, in the AVAiL, JO19907, PRONOUNCE, and ERACLE trials, BC was not observed to have significant OS benefit compared with CT alone [12,13,14,15,16]. Therefore, the position of BC as the first-line treatment in the advanced non-squamous NSCLC is now facing challenges.

The importance of immunotherapy in the treatment of NSCLC has put great significance on some previous studies. In addition, the discussion on how ICIs should be added to the regimen was discussed and a few trials are ongoing to evaluate the efficacy and safety profiles of different interventions. This study confirmed that the use of immunotherapy in driver-gene-negative advanced non-squamous NSCLC would significantly benefit the outcome in terms of survival, which was consistent with several previous studies [17,18]. Based on the Checkmate 227 part one B trial, KEYNOTE-021G trial, and KEYNOTE-189 trials, the combination of nivolumab + CT and pembrolizumab + CT were approved to significantly improve the PFS and OS in patients with driver-gene-negative advanced non-squamous NSCLC [19,20,21,22]. At the same time, three PD-1 inhibitors (carrelizumab, sintilizumab, and tislelizumab) produced in China combined with CT have achieved similar favorable results [23,24,25,26]. In addition to PD-1 inhibitors, PD-L1 inhibitors (atezolizumab and sugemalimab) combined with CT also showed amazing results in the treatment of patients with driver-gene-negative advanced non-squamous NSCLC [27,28,29]. The preliminary results in terms of DIC and DI, nivolumab combined with ipilimumab, with or without CT, subsequently have been considered the most effective novel dual immunological applicability to treat NSCLC [19,30,31]. Since the data of the non-squamous NSCLC subgroup were not provided in the MYSTIC, CHOICE-01, Empower-Lung 3, and POSEIDON studies, these four studies were not included in this NMA [32,33,34,35]. There was no significant difference in OS and PFS among the four ICI-based therapies (BIC, IC, DI, and DIC). However, the improvement in ORR was statistically significant in BIC vs. DIC and IC vs. DIC. One reason may be related to the relatively small number of research studies that met the inclusion criteria of this meta-analysis on DI, DIC, and BIC.

There was not enough evidence for a direct comparison between IC and BC. The only RCT we were able to find with sufficient data was IMpower150, a multicenter, randomized, phase three trial which evaluated the efficacy of first-line atezolizumab + BEV + CT (ABCP) in patients with metastatic non-squamous NSCLC in contrast to atezolizumab + CT (ACP) and BEV + CT (BCP) [36]. In the 4-year updated results of OS in the IMpower150 study, patients with metastatic non-squamous NSCLC in the ACP group versus the BCP group were numerical, but not statistically significant (HR, 0.84; 95% CI: 0.71–1.00). Unfortunately, this study did not report the complete results of PFS, ORR, or ≥3 TRAEs between ACP and BCP subgroups for comparison. In our study, IC appeared to have longer OS (HR, 0.80; 95% CI: 0.67–0.95) and PFS (HR, 0.68; 95% CI: 0.53–0.86) compared with BC, with a similar ORR and ≥3 TRAEs incidence (RR, 0.98; 95% CI: 0.79–1.21/RR, 0.89; 95% CI: 0.61–1.28; respectively) compared with BC, which suggested that IC was a better choice as first-line treatment in driver-gene-negative advanced non-squamous NSCLC (Figure 3). In the SURCA rank, treatments with the probability of being ranked first in OS are as follows: DIC (0.44), BIC (0.33), IC (0.14), DI (0.09), BC (0.00), and CT (0.00) (Appendix A). Subgroup analyses showed that IC yielded superior PFS (HR, 1.59; 95% CI: 1.05–2.38), but not OS than BC in the subgroup of patients < 65 years old. In ECOG, smoking, gender, age, patients ≥ 65 years old, or liver metastases subgroups, no statistical differences were observed between IC and BC (Figure 4). In the SURCA rank of subgroups, an improved OS of patients who were treated with IC compared to patients who were treated with BC was observed in the ECOG PS = 0/1, smokers/non-smokers, and patients < 65 years old, and female subgroups. An improved OS of patients who were treated with BC compared to patients who were treated with IC was observed in patients ≥ 65 years old, male, and patients with liver metastases subgroups (Appendix A). Although when SUCRA prediction contradicts NMA results, the HR estimation of NMA should be given priority due to the non-absolute prediction of SUCRA for treatment strategy ranking, the SURCA ranking results could also serve as good references and help us to screen for patients with driver-gene-negative advanced non-squamous NSCLC who are most likely to benefit from IC or BC.

There were several limitations of this NMA. First, due to the time span of this NMA of more than 10 years, the expression of PD-L1 and tumor mutational burden were not reported in earlier studies (e.g., ECOG 4599, AVAiL, JO19907, and ERACLE trials), which might lead to a partially-biased conclusion. Second, of the 19 studies we included, none of the trials containing IC provided detailed data of OS of patients with a different stage (stage III B or stage IV), so we did not compare the survival outcomes of IC or BC in patients with different stages. Thirdly, different kinds of immunotherapeutic agents in combination with CT or BEV were included in this NMA, which might affect the outcome. Fourth, not all treatment modalities were compared in the subgroup analysis due to the limited data of subgroup analysis in some RCTs. Real-world prospective studies are warranted to validate the reliability of this conclusion.

## 5. Conclusions

In conclusion, this NMA suggested that IC is a better efficient first-line therapy for patients with driver-gene-negative non-squamous advanced NSCLC, with prolonged PFS and OS and comparatively lower risk of ≥3 TRAEs in comparison to BC.

## Figures and Tables

**Figure 1 jcm-11-01655-f001:**
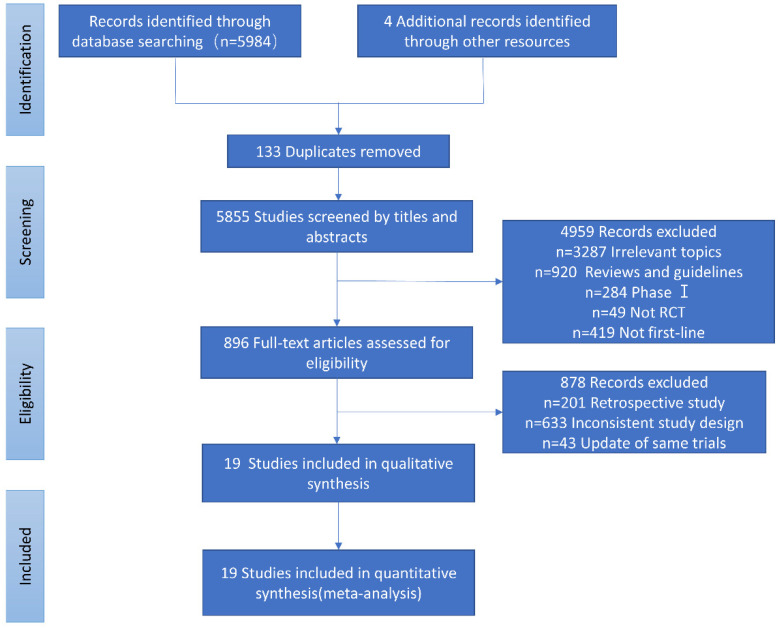
Flow-chart of the literature search.

**Figure 2 jcm-11-01655-f002:**
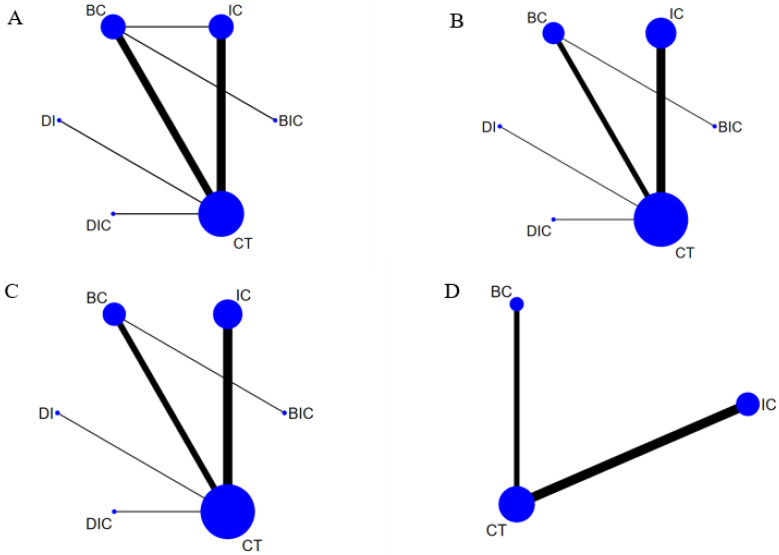
Network plot of OS (**A**), PFS (**B**), ORR (**C**) and ≥3 TRAEs (**D**). Each circular node represents a treatment type. The circle size is proportional to the total number of patients. The width of the line is proportional to the number of studies performing head-to-head comparisons in the same study, and the dotted line is the indirect comparison which was shown in this network meta-analysis.

**Figure 3 jcm-11-01655-f003:**
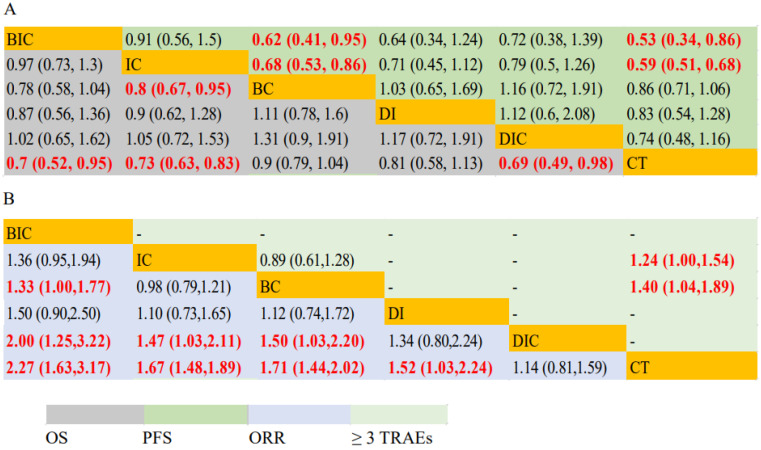
ORR, PFS, OS and safety profiles in the whole cohort analysis according to network meta-analysis. Each cell contains the pooled HR and 95% CI for PFS and OS (**A**) and RR and 95% CI for ORR and ≥3 TRAEs (**B**); significant results are indicated in red.

**Figure 4 jcm-11-01655-f004:**
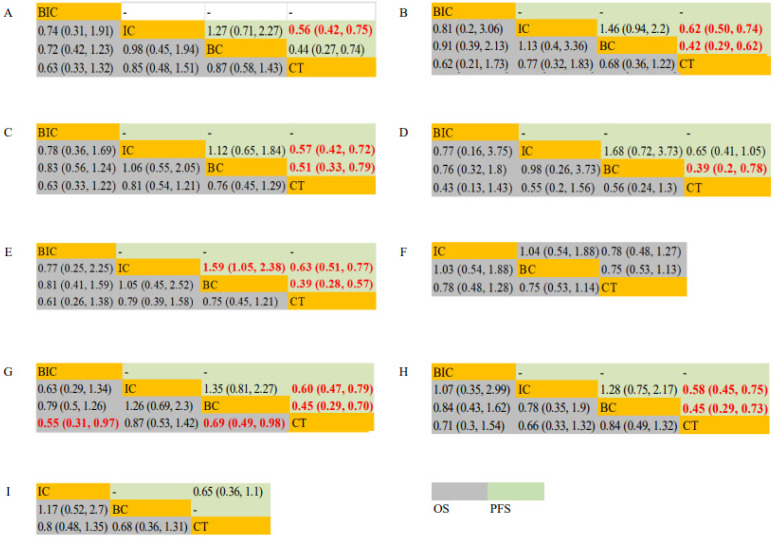
OS and PFS for study cohort in the subgroup analysis based on ECOG = 0 (**A**), ECOG = 1 (**B**), smokers (**C**), non-smokers (**D**), age < 65 (**E**), age ≥ 65 (**F**), male (**G**), female (**H**) and liver metastasis (**I**) according to network meta-analysis. Each cell contains the pooled Hazard Radio (HR) and 95% credibility intervals for OS and PFS; significant results are indicated in red.

**Table 1 jcm-11-01655-t001:** Baseline Characteristics of the included randomized controlled trials.

	NCT Identifier Number	Published Year	First Author	Phase	Arm	Non-Squamous Patients with Survival Data	HR OS (95% CIs)	HR PFS (95% CIs)	ORR (n/N *)	≥3 TRAEs (n/N *)
ECOG4599	NCT00021060	2006	Sandler A, et al.	III	BCP	417	0.79 (0.67 to 0.92)	0.66 (0.57 to 0.77)	133/381	251/427
					CP	433	-	-	59/392	104/440
AVAiL	NCT00806923	2010	Reck M, et al.	III	BCG	351	1.03 (0.86 to 1.23)	0.85 (0.73 to 1.00)	121/351	274/329
					CG	347	-	-	75/347	252/326
JO19907	CTI-060338	2012	Niho S, et al.	II	BCP	117	0.99 (0.65 to 1.50)	0.61 (0.42 to 0.89)	71/121	NA
					CP	58	-	-	18/58	NA
PRONOUNCE	NCT00948675	2015	Zinner RG, et al.	III	BCP	182	1.07 (0.83 to 1.36)	1.06 (0.84 to 1.35)	49/179	NA
					Pem + Cb	179	-	-	43/182	NA
BEYOND	NCT01364012	2015	Zhou C, et al.	III	BCP	138	0.68 (0.50 to 0.93)	0.40 (0.29 to 0.54)	75/138	94/140
					CP	138	-	-	36/138	83/134
ERACLE	NCT01303926	2015	Galetta D, et al.	III	Cisplatin/Pemetrexed	60	0.93 (0.60 to 1.42)	0.79 (0.53 to 1.17)	24/60	6/58
					BCP	58	-	-	30/58	5/60
KEYNOTE-021G	NCT02039674	2016	Langer C, et al.	II	Pembrolizumab + Pem + Cb	60	0.71 (0.45 to 1.12)	0.54 (0.35 to 0.83)	35/60	23/60
					Pem + Cb	63	-	-	21/63	20/63
IMpower130	NCT02367781	2019	West H, et al.	III	Atezolizumab + Nab-paclitaxel + carboplatin	451	0.79 (0.64 to 0.98)	0.64 (0.54 to 0.77)	220/447	381/473
					Nab-paclitaxel+ carboplatin	228	-	-	72/226	164/232
KEYNOTE-189	NCT02578680	2020	Gadgeel S, et al.	III	Pembrolizumab + Pemetrexed + Platinum	410	0.56 (0.45 to 0.70)	0.48 (0.40 to 0.58)	197/410	291/410
					Pemetrexed + Platinum	206	-	-	40/206	135/206
CheckMate 227 4year part1A ^i^	NCT02477826	2020	Hellmann MD, et al.	III	Nivolumab + Ipilimumab	278	0.81 (0.67 to 0.99)	0.83 (0.68 to 1.01)	103/278	NA
					Chemotherapy	279	-	-	91/279	NA
CheckMate 227 4year part1B ^ii^	NCT02477826	2020	Hellmann MD, et al.	III	Nivolumab + Ipilimumab	134	0.69 (0.53 to 0.89)	0.66 (0.51 to 0.87)	53/134	NA
					Chemotherapy	140	-	-	30/140	NA
CheckMate 227 4year part2 ^iii^	NCT02477826	2020	Hellmann MD, et al.	III	Nivolumab + chemotherapy	270	0.69 (0.50 to 0.97)	0.69 (0.50 to 0.97)	130/270	NA
					Chemotherapy	273	-	-	80/273	NA
IMpower132	NCT02657434	2021	Nishio M, et al.	III	APP	292	0.86 (0.71 to 1.06)	0.60 (0.49 to 0.72)	137/292	159/292
					PP	286	-	-	92/286	115/286
IMpower150 4 year update	NCT02366143	2021	Socinski MA, et al.	III	ABCP	350	0.80 (0.67 to 0.95) ^iv^	0.62 (0.52 to 0.74)	224/350	NA
					ACP	359	0.84(0.71 to 1.00) ^v^	NA	NA	NA
					BCP	338	-	-	159/331	NA
CheckMate 9LA 2-year update	NCT03215706	2021	Reck M, et al.	III	Nivolumab + ipilimumab + platinum-doublet Chemotherapy	248	0.69 (0.55 to 0.87)	0.74 (0.60 to 0.92)	82/246	NA
					Chemotherapy	247	-	-	54/246	NA
CameL	NCT03134872	2021	Zhou, et al.	III	Camrelizumab + Pem + Cb	205	0.73 (0.53 to 1.02)	0.60 (0.45 to 0.79)	123/205	NA
					Pem + Cb	207	-	-	81/207	NA
RATIONALE 304	NCT03663205	2021	Lu S, et al.	III	Tislelizumab + chemotherapy	222	NA	0.645 (0.46 to 0.902)	128/223	141/205
					Chemotherapy	110	-	-	41/111	97/207
ORIENT 11	NCT03607539	2021	Yang Y, et al.	III	Sintilimab + pemetrexed + platinum	266	NA	0.49 (0.38 to 0.63)	138/266	74/223
					Pemetrexed + platinum	131	-	-	39/131	23/111
GEMSTONE-302	NCT03789604	2021	Zhou C, et al.	III	Sugemalimab + platinum-based chemotherapy	191	NA	0.59 (0.45 to 0.79)	NA	NA
					Platinum-based chemotherapy	96	NA	-	NA	NA

Remarks: N *: Patients with reported results of ORR or ≥3 TRAEs; ^i^: nivolumab plus ipilimumab versus chemotherapy across subgroups in patients with tumor PD-L1 expression ≥ 1%; ^ii^: nivolumab plus ipilimumab versus chemotherapy across subgroups in patients with tumor PD-L1 expression < 1%; ^iii^: nivolumab plus chemotherapy versus chemotherapy in patients with non-squamous NSCLC; ^iv^: ABCP versus BCP in ITT-WT patients; ^v^: ACP versus BCP in ITT-WT patients. Abbreviations: ORR: overall response rate; TRAEs: treatment-related adverse events; CP: paclitaxel and carboplatin; ABCP: atezolizumab + bevacizumab + paclitaxel + carboplatin; ACP: atezolizumab + paclitaxel + carboplatin; BCP: paclitaxel and carboplatin plus bevacizumab; CG: cisplatin + gemcitabine; BCG: bevacizumab + cisplatin + gemcitabine; Pem + Cb: pemetrexed + carboplatin; APP: atezolizumab + cisplatin/carboplatin + pemetrexed; PP: cisplatin/carboplatin + pemetrexed; OS: overall survival; PFS: progression-free survival; HR: hazard ratio; CI: confidence interval; NSCLC: non-small cell lung cancer; ITT-WT: intention-to-treat wild-type; PD-L1: programmed death ligand-1; NA: not available.

## Data Availability

The datasets developed and analyzed during this study are available from the corresponding author upon reasonable request.

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
