# Peer review of "Combined Immunotherapy with Chemotherapy versus Bevacizumab with Chemotherapy in First-Line Treatment of Driver-Gene-Negative Non-Squamous Non-Small Cell Lung Cancer: An Updated Systematic Review and Network Meta-Analysis"

_jcm, 2022, doi:10.3390/jcm11061655_

Round 1

Reviewer 1 Report

I found your work well organized well implemented and well written. Network metanalysis is inherently a technique that provides high levels of uncertainty as it uses indirect comparisons and the fact that different chemotherapy regimens were used for the indirect comparisons makes matters worse, however, you already mention this in your limitations.

My only comment would be that the article does not provide much new knowledge, since immunotherapy for driver-gene-negative NSCLC is already here and has been here for quite sometime in the 1st line.

Author Response

Response: Thank you for your valuable suggestion. Currently, the only one RCT (IMpower150) we are able to find with sufficient data directly compared the efficacy between IC and BC. However, IMpower150 mainly explored the efficacy between ABCP vs. ACP and ABCP vs. BCP, and did not provide detailed data on PFS and adverse events between ACP and BCP. Second, IMpower150 mainly found that for the patients with PD-L1 expression ≥ 1%, the OS of ACP (IC) was better than BCP (BC). However, few studies directly compared the efficacy and safety of CT with immunotherapy or anti-angiogenetic agents in the first-line treatment for patients with driver-gene-negative advanced non-squamous NSCLC. Thirdly, the number of cases for comparing the efficacy between IC and BC for patients with NSCLC is relatively limited, so it is very necessary to conduct an NMA to explore which of IC and BC has better efficacy and lower toxicity. These outcomes will provide evidence for the selection of rational therapeutic regimens in the clinic.

Reviewer 2 Report

Chai et al. propose a review with a meta-analysis conducted to evaluate the efficacy of different first line therapies in NSCLC.

Lung cancer is the leading cause of cancer-related mortality. Because treatment is not satisfactory for almost all patients with NSCLC, any study focused on this pathology has great clinical and social relevance. An exhaustive review of the current state, as proposed by the authors, may be helpful to the medical community.

 The strong point of the article is undoubtedly the massive bibliography search.

The authors dissect in details the effects of IC and BC uncovering PFS and OS in all patients enrolled and the conclusion is clear and concise.

The limitations of the study are also listed in the paper and they are mainly due to the variety of immunotherapy agents used in the different trials and the lack of an adequate randomized control group. Also, the fact that trials from more than ten years ago are included in the study, must be considered, taking into account the analytical and computer techniques at the time.

Roughly, I consider this is a complete and current review that meets all the criteria for publication in JCM.

I would just like to refer to some minor suggestions:

  • Line 20, non-squamous must be removed as NSCLC is detailed right after
  • Figure 2 : letters in the figure are not visible at all
  • Figure 4 has different format (font and size) and a worse quality that Figure 3
  • Do the authors know the stage of the NSCLC tumours analysed? They mentioned “patients with driver-gene-negative advanced NSCLC”, are all of them stage III or IV? Could this fact affect to the results?
  • To me, Figure 3 and 4 are difficult to read and understand. A new graph indicating only the significant results (numbers in red) could help.
  • Regarding the same figures, have the authors tried a forest plot to facilitate the viewing of the results?

Author Response

  1. Comment 1:

Line 20, non-squamous must be removed as NSCLC is detailed right after

Figure 2 : letters in the figure are not visible at all

Figure 4 has different format (font and size) and a worse quality that Figure 3

Response: We agree with your suggestion. Non-squamous has been removed in Line 20. Figure 2 and Figure 4 have been redrawn.

  1. Comment 2:

 Do the authors know the stage of the NSCLC tumours analysed? They mentioned “patients with driver-gene-negative advanced NSCLC”, are all of them stage III or IV? Could this fact affect to the results?

Response: We agree with your comment. All of the patients with driver-gene-negative advanced NSCLC enrolled in this study were stage III B or IV, which has been mentioned in the inclusion criteria. Of the 19 studies we included, none of the trials containing immunotherapy provide detailed data of OS of patients with a different stage (stage III B or stage IV), so we did not compare the survival outcomes of IC or BC in patients with different stages (stage III  B or stage IV). Further study is warranted to further explore whether this factor affects the results. And this limitation of this study has been added in the Discussion part. (page 11, lines 346-348)

  1. Comment 3: To me, Figure 3 and 4 are difficult to read and understand. A new graph indicating only the significant results (numbers in red) could help.

Regarding the same figures, have the authors tried a forest plot to facilitate the viewing of the results?

Response: Thank you for this valuable suggestion. Forest plots of hazard ratios (HRs) for OS and PFS,  risk ratios (RRs) for ORR and TRAE in NMA are presented in Supplemental Figures 2, 3, 4, 5, which may be helpful to read and understand the results. But league charts (Figure 3 and Figure 4) are a simpler, more concise, and more aesthetically pleasing way to display NMA results, which have been used in most NMAs currently. Therefore, we still choose the league chart as the main chart of the conclusion and put it in the main text. All positive results are marked (red, bold) in the figures and explained in detail in the body of the manuscript (Results part). Your valuable opinions made this article get significant improvement. Thank you very much again!

Reviewer 3 Report

The present meta-analysis revealed that immunotherapy plus chemotherapy (IC) had significantly prolonged overall survival (OS) and progression-free survival (PFS) compared with other therapies to non-small cell lung cancer (NSCLC) including bevacizumab plus chemotherapy. Clear analytic processes and data expressions seems to be adequately described in the present systematic review. Below minor comments may be considered.

In the Introduction, there are no information regarding the mutations of indicated driver genes in NSCLC. Several driver genes like EGFR, HER2, RET, and MET should be described for further understanding on driver-gene-negative NSCLC.

Author Response

Response: Thank you for your valuable suggestion. Common driver genes of NSCLC included EGFR, ALK, ROS1, BRAF, MET (exon 14 skipping mutation), NTRK, RET, etc, which has been defined in the National Comprehensive Cancer Network (NCCN) guidelines of NSCLC in 2021. The most prominent driver gene mutations were sensitive epidermal growth factor receptor (EGFR) mutations, followed by anaplastic lymphoma kinase (ALK) rearrangements. Driver-gene-negative NSCLC in this NMA refers to NSCLC patients with no sensitive EGFR mutations or ALK rearrangements according to the 19 enrolled studies cause nearly all the trials enrolled in this study were conducted before 2021 and only evaluated the presence of EGFR mutation and ALK status (page 3, line 96). Based on your suggestion, we have added the description of driver-gene-negative in this article (page 1, lines 35-41).